# 'They've all endorsed it…but I'm just not there:' a qualitative exploration of COVID-19 vaccine hesitancy reported by Black and Latinx individuals

David Scales ,[1,2] Sara Gorman,[2] Savannah Windham,[2] William Sandy,[3] Nellie Gregorian,[3] Lindsay Hurth,[2] Malavika Radhakrishnan,[2] Azubuike Akunne,[2] Jack M Gorman[2]

¹Division of General Internal Medicine, Weill Cornell Medical College, New York, New York, USA
²Critica, The Bronx, New York, USA
³Fluent Research, New York, New York, USA

**Correspondence to**
Dr David Scales;
david.scales@aya.yale.edu

## ABSTRACT

**Objective** We sought to examine reasons for vaccine hesitancy among online communities of US-based Black and Latinx communities to understand the role of historical racism, present-day structural racism, medical mistrust and individual concerns about vaccine safety and efficacy.
**Design** A qualitative study using narrative and interpretive phenomenological analysis of online bulletin board focus groups.
**Setting** Bulletin boards with a focus-group-like setting in an online, private, chat-room-like environment.
**Participants** Self-described vaccine hesitant participants from US-based Black (30) and Latinx (30) communities designed to reflect various axes of diversity within these respective communities in the US context.
**Results** Bulletin board discussions covered a range of topics related to COVID-19 vaccination. COVID-19 vaccine hesitant participants expressed fears about vaccine safety and doubts about vaccine efficacy. Elements of structural racism were cited in both groups as affecting populations but not playing a role in individual vaccine decisions. Historical racism was infrequently cited as a reason for vaccine hesitancy. Individualised fears and doubts about COVID-19 (short-term and long-term) safety and efficacy dominated these bulletin board discussions. Community benefits of vaccination were not commonly raised among participants.
**Conclusions** While this suggests that addressing individually focused fear and doubts are central to overcoming COVID-19 vaccine hesitancy in Black and Latinx groups, addressing the effects of present-day structural racism through a focus on community protection may also be important.

## STRENGTHS AND LIMITATIONS OF THIS STUDY

⇒ We employed online bulletin board groups of US-based Black and Latinx participants selected for self-described vaccine hesitancy to understand the reasons and motivations behind their stance, leveraging the online design to attract participants whose information environments consist of high proportions of online content where anti-vaccine misinformation was prominent.

⇒ Compared with in-person focus groups, asynchronous bulletin boards allow all participants to freely express themselves with fewer concerns about speaking too much while facilitators can encourage greater participation from those who are more quiet.

⇒ Transcripts were analysed using various qualitative techniques, including narrative and interpretive phenomenological analysis to allow for understanding recurrent themes.

⇒ Participants revealed that vaccine hesitancy is the result of a confluence of psychological and social considerations, but with selective focus on certain factors over others as participants weighed risks and benefits, such as high emphasis was placed on individual vaccine safety with relatively little attention to potential community-level benefits of vaccination.

⇒ Vaccine safety and efficacy were of highest concern; however, mistrust in institutions and concerns about systemic and personal racism also featured prominently among participants' concerns.

## INTRODUCTION

For much of the COVID-19 pandemic, rates of COVID-19 vaccination in Black and Latinx communities in the USA were lower than white communities, although the gap appears to be narrowing.[1–3] This vaccination gap is especially concerning because Black and Latinx people diagnosed with COVID-19 have experienced worse clinical outcomes.[4]

Structural and social determinants of health such as racism, socioeconomic status, access to transportation and access to information or trusted healthcare practitioners make it difficult for people in some communities who want to be vaccinated to get vaccines, but even as vaccine uptake gaps have narrowed, a number of people continue to choose not to be vaccinated.[5–7]

COVID-19 vaccine hesitancy defined as 'indecision around accepting a vaccination',[8]

among Black and Latinx people has been found in survey studies to be higher than among white people.[9–11] In their review of 13 studies of racial and ethnic disparities in COVID-19 vaccination status, Khubchandani and Macias[9] found an overall pooled rate of vaccine hesitancy of 26.3%, but a higher rate among Hispanic (30.2%) and African American (40.6%) study participants.[12] Even among healthcare workers, COVID-19 vaccine hesitancy was found to be higher among Black and Latinx people compared with white people.[13]

The literature on COVID-19 vaccine hesitancy among Black and Latinx people highlights three influential factors affecting vaccine decisions in these populations: individual fears and concerns about vaccine adverse side effects and efficacy, historical mistreatment in medical/scientific contexts (eg, the US Public Health Service syphilis study at Tuskegee), present-day experience with mistreatment by the healthcare system, (including structural racism), the latter two of which are linked to mistrust in healthcare providers and the healthcare system. The literature on each of these topics is vast and a full review is beyond the scope of our study. However, there are some key studies worth highlighting that informed and motivated the research presented here.

Survey data have shown that concerns about vaccine safety and efficacy are associated with higher rates of vaccine hesitancy among Black and Latinx people.[14 15] In a survey using US census data, Black people were more likely than white people to develop COVID-19 vaccine hesitancy because of lack of confidence in the safety and efficacy of vaccines and because of a tendency to watch evolving information and wait before considering vaccination, though this group saw the greatest percentage drop in hesitancy over time.[16] Longoria *et al* found fears about COVID-19 vaccine safety were commonly circulated online among Latinx people, as well as narratives about alleged 'alternative treatments'.[17] Recent work by Morales and Paat provides additional evidence of a 'watch-and-wait'" approach among Black Americans, noting how rates of vaccine hesitancy and refusal in this community declined over time while it remained stable in white communities,[18] consistent with more people seeking the vaccine as more time passes demonstrating the safety and efficacy of the vaccine.

Early in the vaccine roll-out, there was significant concern among public health professionals that well-known narratives of historical racism among marginalised communities around, for example, Henrietta Lacks[19] or the syphilis study at Tuskegee,[20] would engender vaccine hesitancy due to a 'legacy of distrust' in medical research.[21] However, much empirical research highlighted that past narratives around Henrietta Lacks or the unethical syphilis experiments performed in Tuskegee relate to and shape contemporary lived experiences, perceptions of racism in medicine, and mistrust in the healthcare system in general and COVID-19 vaccines specifically.[22] Using survey data, Martin *et al*[23] found that current mistreatment by the healthcare system, rather than historical

mistreatment as exemplified by the Tuskegee experiments, was associated with COVID-19 vaccine hesitancy among Black Americans. Another small study among Black undergraduates described these historical examples of racist and unethical practices as 'backdrops' that informed their contemporary perceptions of how Black people continue to be discounted.[24]

Additionally, present-day manifestations of structural racism with historical antecedents, for example, the downstream impact of redlining, perpetuate healthcare access and socioeconomic disparities[25 26] that subsequently are likely to influence vaccine-related decisions. For various geographical and socioeconomic reasons, Black people are less likely to have access to a primary care physician[27–29] and more likely to use emergency care, a relationship partly mediated by mistrust in the healthcare system.[28] Therefore, while primary care physicians are often cited across different racial and ethnic groups as the most trusted person when it comes to vaccine decision-making,[30 31] structural inequalities in healthcare access means that many marginalised communities lack access to these heavily trusted sources.[32 33] Some population-based studies support the link between access to primary care and those provider recommending vaccines to higher rates of COVID-19 vaccine uptake.[34 35]

As alluded to above, the lived experience of historical racism, present-day racism, including structural manifestations of it, and medical mistrust are intertwined. For example, one qualitative study linked historical mistrust to the way Black people in the Deep South have been treated by the healthcare system as a factor contributing to vaccine hesitancy.[36] Another focus group study with Black and Latinx community members also identified 'pervasive mistreatment' as a basis for vaccine hesitancy in those communities,[37] suggesting that it is difficult to separate perceptions of medical racism from institutional mistrust in healthcare. Similarly, an analysis of online posts found that mistrust of vaccines and the motivations of official institutions (ie, institutional mistrust) were commonly expressed in online platforms viewed by the Black community.[38] Similar concerns were heard in focus groups of Black salon and barbershop owners[39] and in a study focusing on older Black and Hispanic adults.[40] In addition to these fears, Bateman *et al* conducted virtual focus groups and identified mistrust of the COVID-19 vaccine development process among Black and Latinx participants from the Deep South.[36] Similarly, while conceptually separated here, concerns about vaccine safety are also likely intermixed with institutional mistrust and experiences with racism since safety is assessed by government agencies. For example, a survey of people in underserved communities in North Carolina identified safety concerns and government mistrust as the most important factors for vaccine hesitancy among Black and Latinx respondents.[41]

The purpose of this study was to probe more deeply into these factors to get a better understanding of the complexities involved in lower COVID-19 vaccine uptake

among Black and Latinx people. To do this, we conducted two separate online bulletin boards, one with participants from each community who self-identified as vaccine hesitant, to probe the factors behind their vaccine decisions. We were interested in understanding what motivates people in the Black and Latinx communities to be COVID-19 vaccine hesitant. Originally, we intended this as two separate inquiries, one involving Black and the other involving Latinx participants and therefore the designs and recruitment strategies of the online bulletin boards differed. However, we observed remarkably similar responses from participants in the two groups and therefore decided to combine them into a single report.

These bulletin boards were conducted within the first few months after COVID-19 vaccines were made available and reflect attitudes at that time. Gaps in COVID-19 vaccination rates among racial and ethnic groups have since narrowed and attitudes about vaccines may have shifted. However, the results of these bulletin boards remain important for two reasons: first, because they provide insight into important drivers of vaccine hesitancy in Black and Latinx communities; and second, because they may help inform strategies to support future vaccine demand as new healthcare challenges inevitably arise.

## METHODS

We conducted two bulletin boards from 13 July 2021 to 22 July 2021, following the COVID-19 vaccine roll-out in the USA Informed consent via an online form was obtained from each participant prior to the start of the study, and they were assured that participation was voluntary. Participants were told that they could end their participation at any time and were free to leave any questions unanswered. Subjects were paid US$120 for their participation.

A bulletin board is an asynchronous online discussion involving greater numbers of individuals than typical focus groups and taking place over an extended period.[42–44] Participants log into a password-protected site run by an external third party (QualBoard, since acquired by Sago) that creates such dedicated platforms to answer questions that are posted and monitored by a moderator. The moderator can also follow up on responses for clarification or elaboration. The bulletin board is a flexible research tool that allows the moderator to post questions and probe any individual participant following their entry. The respondents can take as much time to respond as they need. Individual responses are initially uninfluenced by the group, as participants do not see other responses to any given question until they have posted their own response. This method helps to minimise the social desirability bias[45] that may influence participants after exposure to another's responses.

### Data collection

Participants in the bulletin board with Black participants were recruited from a panel of people who have previously agreed to participate in online surveys. They were contacted by email with an invitation to participate. If interested, they were asked to respond to an online screener that assessed their level of vaccine hesitancy. People who had already been vaccinated, intended to be vaccinated soon or who adamantly opposed COVID-19 vaccination were excluded. For recruiting purposes, we defined vaccine hesitancy as adults who were not categorically opposed to vaccines, but were undecided as to whether it was safe to receive the new COVID-19 vaccines. We then screened for individuals that met these criteria. If qualified, they provided their contact information and were given instructions for logging into the bulletin board. Please see table 1 for demographics of the participants.

The process for recruiting participants from Latinx communities differed from that for Black participants. We posted invitations in Spanish on various Facebook pages created for Latinx subpopulations, such as groups for communities from Peru, Colombia, Mexico and the Dominican Republic. If interested, they were asked to complete the online screening questionnaire in Spanish, to determine if they met the criteria for participation, which were the same as for the participants in the Black groups as described above. If qualified, they provided their contact information and were supplied with instructions for logging into the bulletin board. This project was designed to inform subsequent interventions to address vaccine hesitancy. Therefore, the demographics of each group were chosen to approximately match those of the groups in which interventions would take place in a later study based on observational assessment of such spaces by our interventionists (for more info on these interventions, see references 46 47). Similarly, our funding for this project came from a source exclusively prioritising health in the USA, we focused on finding participants that were relatively representative of these racial and ethnic groups living in the USA according to US Census American Community Survey 5-year estimates.

When participants logged into the bulletin board, they were presented with an introduction from the moderator, a review of the process, and a reminder that they were not obliged to answer any question. They were reassured that the research was anonymous and their identities, including contact information, would not be shared. The moderators of the bulletin boards introduced themselves at the outset and posted their photographs so that the respondents could see them. Participants were allowed to post photographs of themselves to the group, though this was not required.

Participants were then presented with the first of a series of questions. Only after a participant entered their response to a question were they able to see how other participants responded to that same question. At this point, they were free to respond to what other participants had said. After responding to all of the questions posted for that day, they were reminded to check back periodically to respond to possible follow-up questions posted by the moderator. This process continued over 3 days, with a different set of questions posted each day.

**Table 1** Characteristics of bulletin board participants

| | Participants from the Black community (N=30) | Participants from the Latinx community (N=30) |
|---|---|---|
| Gender | | |
| Male | 9 | 14 |
| Female | 21 | 16 |
| Age | | |
| 18–29 | 4 | 13 |
| 30–39 | 8 | 8 |
| 40–49 | 7 | 6 |
| 50–59 | 6 | 3 |
| 60–69 | 5 | 0 |
| Marital status | | |
| Married or living with partner | 5 | 20 |
| Divorced or widowed | 6 | 0 |
| Single | 19 | 10 |
| Education level | | |
| Less than high school | 1 | 3 |
| Some college | 15 | 13 |
| College degree | 10 | 10 |
| Postgraduate | 4 | 4 |
| Household income | | |
| Below US$35 000 | 9 | 7 |
| US$35 000–US$49 999 | 5 | 3 |
| US$50–US$74 999 | 5 | 11 |
| US$75–US$99 999 | 8 | 10 |
| US$100 000+ | 3 | 0 |
| Influenza vaccine history | | |
| Usually get the vaccine | 4 | 3 |
| Sometimes get the vaccine | 12 | 13 |
| Never get the vaccine | 14 | 14 |
| Religion | | |
| Roman catholic | NA | 19 |
| Protestant | NA | 7 |
| None | NA | 3 |
| Mormon | NA | 1 |
| US or foreign-born | | |
| US-born | NA | 17 |
| Foreign-born | NA | 13 |
| Heritage country | | |
| Mexico | NA | 8 |
| Peru | NA | 4 |
| Ecuador | NA | 4 |
| Dominican Republic | NA | 3 |
| Venezuela | NA | 2 |

Continued

**Table 1** Continued

| | Participants from the Black community (N=30) | Participants from the Latinx community (N=30) |
|---|---|---|
| Puerto Rico | NA | 2 |
| Colombia | NA | 2 |
| El Salvador | NA | 1 |
| Chile | NA | 1 |
| Costa Rica | NA | 1 |
| Cuba | NA | 1 |
| Guatemala | NA | 1 |

NA, not applicable.

Bulletin board questions were designed for flexible, open-ended inquiry. The research did not seek to confirm any hypotheses but rather to explore the range of perceptions and attitudes that exist in the vaccine hesitant population and to identify important influencers of those perceptions and attitudes, including trusted sources of information, media outlets, social networks, community leaders, health professionals, etc. Examples of the various topics of inquiry and discussion can be found in table 2 and the facilitators guide included in online supplemental material. We also asked participants' perspectives on influenza vaccines, but only data from questions about COVID-19 vaccines are included here for the following reasons.

First, we found participants more expressive about COVID-19 vaccines and terse in their feelings about influenza vaccination by comparison. Our resulting analysis of influenza discussions quickly reached thematic saturation, with less range of sentiment than what has been reported elsewhere in the literature.[48 49] Attempts to probe this lack of interest/enthusiasm in influenza vaccination were unsuccessful.

### Data quality control
The study employed purposive sampling with screening to ensure that respondents reflected the target population in terms of attitudinal, behavioural and demographic characteristics. The sample was highly diverse with respect to age, geography, socioeconomic status and in the case of the Latinx sample, with respect to both level of acculturation to the US and national heritage (see table 1).

The bulletin boards were conducted by trained moderators, each with 20+ years of qualitative research experience. The Latinx bulletin boards were conducted in Spanish by a Latinx moderator; the Black bulletin boards were conducted in English by a Black moderator. The Spanish-language discussion among Latinx respondents was translated into English by an automated translation programme provided by the online platform. This was done for the benefit of those observing the discussion

**Table 2** Topics of inquiry and discussion on the bulletin board

| Day | Topics |
|---|---|
| 1 | General health and well-being concerns for themselves and their families<br>Sources of health and medical information and advice<br>Primary care doctors<br>Use of trusted family and home remedies<br>Personal experiences with vaccines in the past<br>Experiences with influenza vaccines<br>Awareness of messaging around vaccine safety<br>Preferred sources of information<br>Use of social media for medical or health information |
| 2 | Things they have heard about the COVID-19 vaccines<br>How much they trust the sources<br>What, if anything, frightens them about a COVID-19 vaccine<br>Which is more frightening to them, catching COVID-19 or getting a vaccine<br>Intentions regarding a COVID-19 vaccine<br>Perceived effectiveness of the COVID-19 vaccines<br>If and how they have discussed the vaccine with their doctors<br>If and how they have discussed the vaccine with family members or friends<br>What their community and church leaders are advising them with respect to the vaccine<br>What they have heard about vaccines and vaccine safety on social media<br>How much they trust what they see on social media<br>What public health officials are saying<br>How much trust they place in public health officials |
| 3 | How have their communities and their families been affected by COVID-19<br>How worried are they about possibly passing COVID-19 on to at-risk members of their families<br>How important do they feel it is to eventually receive a COVID-19 vaccine<br>How important are vaccines for restoring normalcy<br>What are the best arguments they have heard in favour of vaccination<br>How do they feel about the idea of mandated vaccination<br>What information would make them feel better about getting a COVID-19 vaccine<br>Whose endorsement of vaccination would be meaningful for them<br>Responses to various provaccine messages |

who were not Spanish speakers. The automated translation was not used, however, for the purposes of reporting due to some translation errors. Transcripts included in the report were translated by professional Spanish-speaking moderators.

## Methods of analysis

A combination of methods was employed in the analysis of the content generated by these bulletin boards, including interpretive phenomenological analysis (IPA), narrative analysis, and, secondarily, qualitative content analysis. Taking a reflexive, phenomenological constructivist approach to understand participant viewpoints, these methods enabled us to explore how respondents narrate and make sense of their prior experiences with vaccines, with medical professionals, and with various sources of medical and health-related information. They also enabled us to observe how participants rationalise their hesitancy with respect to COVID-19 vaccination, and to identify a range of social, emotional and perceptual barriers to vaccination. Analysis enabled us to identify the range of opinions exhibited, opinions that are universally shared and those that are more idiosyncratic and portray how different perceptions tend to be clustered or coupled.

Initial coding was done manually by WS. Transcripts were read through once in their entirety by prior to coding. On a second readthrough, relevant text was highlighted, codes were inductively drawn out, and labelled in text margins. Codes were then aggregated and organised into themes which were discussed in meetings with coauthors discussed in more detail below. Themes were then arranged with key quotations pulled out as illustrative examples.

Data credibility was assessed through discursive triangulation. Initial coding was followed by a process of layered discussions. Specifically, that meant multiple meetings between the primary data analyst and a supervisor to process codes and themes, followed by further coding and thematic discussions and revisions with the first and senior author. After consensus on themes had been reached by those four authors, subsequent review and discussion then took place with the remaining coauthors.

### Interpretive phenomenological analysis

Data collection was not designed to test hypotheses or preconceptions, nor was data analysis. The intent was to use the data gathered to better understand the experiential world of the respondents, how they understand the phenomenon of the ongoing pandemic, and how they rationalise their decision to refrain from vaccination.[50] Through this bottom-up analysis, we sought commonalities and patterns in experiences and shared forms of reasoning to inform a richer understanding of vaccine hesitancy. In addition, the analysis included any consistent variations in participants' responses that corresponded with major demographic variables such as gender and age. For the descriptive analysis, we identified and catalogued the fullest possible range of opinions around vaccine hesitancy, including commonly cited sources of information, facts, anecdotes and trusted sources of information, regarding the pandemic and COVID-19 vaccines.

### Narrative analysis

In addition to identifying and cataloguing the range of opinions and perceptions articulated by participants, the analysis focused on identifying the ways in which information has been woven into narratives. This analysis focused on participants' descriptions of their experiences during

the pandemic, their methods for searching for and processing relevant information, and the stories they tell themselves about the need or lack of need for a vaccine. In addition, we analysed the trajectory of each individual participant's experience with vaccines, looking to identify key moments when their attitudes reportedly changed. This analysis also sought to identify pre-existing narratives and how those intersect with participants' narratives about the pandemic, such as mistreatment of marginalised populations by the healthcare system and lack of trust in the government. This analysis also attempted to gauge the extent to which participants' narratives are fixed, are still being formed, or remain open to revision.

As an additional check on our data and to ensure we met the objective of the study, we examined certain topics of high concern (historical racism, present-day structural racism and medical mistrust) using a directed approach in secondary content analysis.[51]

### Patient and public involvement
None.

### RESULTS
We conducted one bulletin board with 30 people from the Black community and one with 30 people from the Latinx community. Characteristics of the participants can be found in table 1. The themes obtained from the bulletin boards about COVID-19 vaccines in both the Black and Latinx groups were remarkably similar and therefore we combined them in this section. The analysis suggests several interrelated barriers to COVID-19 vaccination are at work in both Black and Latinx communities, strongly influencing vaccine behaviours in these populations. Five main themes and several subthemes emerged. Illustrative quotations can be found in tables 3–5:

1. Safety concerns (table 3).
   - Vaccine unknowns. Vaccines are a 'black box'. Some participants perceived vaccine ingredients to be elusive or intentionally obscured with mysterious ingredients.
   - Fears about COVID-19 vaccine safety. Participants expressed many fears and doubts regarding both the short-term and long-term safety of the vaccines; even those who express high trust in doctors and science and low trust in social media still say stories of vaccine-induced illness make them highly uncertain.
   - Conviction that the COVID-19 vaccine can kill you. Some participants believe that the vaccine is directly responsible for deaths.
   - Concerns about scientific uncertainty. Public scientific debates about vaccine safety and adverse side effects instil and perpetuate doubts by creating the appearance of scientific uncertainty even among those who normally trust medical professionals. Many seem to almost throw up their hands and say, "I can't decide what's true and what's not, so best to

**Table 3** Example quotations of hesitancy related to safety and efficacy concerns

| Subtheme | Quotations |
|---|---|
| Vaccines are a "black box" | "I don't know what they're putting in my body." |
| Fears about vaccine safety | "Vaccines lower the fear of COVID, but not the fear of long-term effects"<br>"The unknown frightens me. What happens when the vaccine interacts with medications… what happens years from now?"<br>"What frightens me is that uncertainty. No one knows what this vaccine will do to humans long term. Let alone babies that are born after." |
| Conviction that the vaccine can kill you | "I believe I would say receiving the vaccine is most frightening. I have had several people to pass away [sic] after receiving the vaccine. Prior to the vaccine these individuals were healthy and doing fine."<br>"The idea that I could die or have health complications because of the vaccine frightens me. I've mostly read this in social media." |
| Concerns about scientific uncertainty | "There are so many conflicting reports that it is difficult to know who is being honest and factual." |
| Vaccines are not effective | "With all the reports of fully vaccinated people contracting COVID a second time I'm not convinced that the vaccine offers the protection it claims."<br>"From what I'm hearing it would be very effective, but some people, even though they got vaccinated ended up with COVID. I'm not really sure at the moment to be honest with you" |
| Vaccines are insufficient | "Currently, only 50% effective. I have seen where there's a booster shot required every six months. I've also heard doctors and CDC state that it doesn't prevent you from getting COVID, it just lessens your likelihood the virus being as bad." |
| Vaccines do not prevent transmission | "Some people that have been vaccinated have gotten the virus. I think that it causes people to lower their guard regarding social distancing and wearing masks. New more contagious strains of the virus are still popping up."<br>"They're dying from the vaccine as well. And the vaccine is not effective. They still get the virus and pass it on to other people." |

**Table 4** Example quotations of hesitancy related to the perception that vaccines are not worth the risk

| Subtheme | Quotations |
|---|---|
| Vaccines are riskier than the virus | *"I have heard the COVID virus isn't too bad and I have multiple friends that have had it. So, I guess you can say I'm more worried about the vaccine than the virus itself."* <br> *"I have already had the virus and had minimal symptoms. So I guess I could say getting the vaccine is more frightening."* <br> *"Both are scary, but getting the vaccine is more frightening for me because I feel that if I got COVID I would be fine and it wouldn't affect me much."* <br> *"At this point, me getting the COVID vaccine is more frightening [than getting COVID). I stay by myself; I only go out if need be and I am masked up."* |
| Vaccines are not necessary | *"I'm not frightened at all because I take great precautions. I'm more concerned about someone passing it on to me."* |
| Vaccines are only for the most vulnerable | *"For me personally, I don't feel like it's necessary as I am a healthy individual with no underlying health issues, and so is my husband and child."* <br> *"I think that the vaccine is important for those who are most vulnerable. If they get sick, at least it won't be as serious."* |

do nothing," or to wait for more conclusive information.

2. Skepticism about vaccine efficacy (table 3).
   – COVID-19 vaccines are not effective. Several stories about new variants, breakthrough infections, and surging cases suggested a belief that the vaccine would not be effective in protecting them.
   – COVID-19 vaccines are insufficient. Even with an effective vaccine, mass vaccination is not enough to return life to normal and that COVID-19 is here to stay, implying that the vaccine's benefits may be exaggerated.
   – COVID-19 vaccines do not prevent transmission. Although vaccines reduce the risk of transmission, news that people can still pass COVID-19 on to others even after being vaccinated is conflated with a narrative that vaccines do not work as intended, thus undermining the argument for getting it to protect others.

3. Risk/benefit calculations were not perceived to favor vaccines (table 4).
   – Vaccines are riskier than the virus. Participants frequently assigned greater risk to the vaccine than to the virus itself and noted that there are other ways to prevent infection (like masking), so on balance the vaccines are felt to be unnecessary.
   – COVID-19 vaccines are not necessary. Participants in both groups often believed they were not at risk of dying from COVID-19; they believed they could contract the virus and recover from it. They also believed that any illness would be mild, underscoring a lack of urgency to be vaccinated.
   – COVID-19 vaccines are only for the most vulnerable. Vaccines are for the most vulnerable, such as older people and immunocompromised people, not the young and healthy, or those being careful and taking other precautions.

4. Limited trust in institutions (table 5).
   – Limited trust in physicians. Many say that they trust their primary care doctors the most when it comes to their health, but that trust does not always extend to advice about the COVID-19 vaccines; they do not necessarily see their doctors as experts in this regard. For instance, some seem to say, 'at this point, no one can claim to be an expert on these vaccines. So, no one can truly tell me what is best'.
   – Lack of trust in government. There is a lack of trust in government in general and especially in government spokespersons, undermining their authority as credible messengers. Many tune them out or do not lend them credence, even those who otherwise trust their doctors and medical professionals. Some people suggest that the very fact that the government so badly wants them to get a vaccine makes them not want to get it.
   – Limited trust in public health authorities. While some of the vaccine hesitant respondents expressed very high regard for medical professionals and for public health authorities in general, they were more critical of public agency-relayed information about COVID-19 vaccines. Some argued that public health authorities only say what they are told to say by the administration. Some participants mistrust the Centers for Disease Control and Prevention (CDC), largely because they viewed the agency as frequently changing its advice and guidelines.

5. How health outcomes differ by race and/or ethnicity (table 5).

Participants in both groups perceived structural racism as factors that influence a group's risk of infection and the likelihood of having access to vaccines. However, the participants did not cite structural issues as influencing either their own personal risk of infection or their decision to be vaccinated. Two people mentioned the Tuskegee experiments[52] as indicative of abuses against Black people by the healthcare system and a reason to be wary of healthcare system programmes, including COVID-19 vaccines. The potential benefits of vaccination, such as protecting vulnerable communities, were not raised as a motivation for vaccination.

### Other observations

Although themes and subthemes about vaccine hesitancy were quite similar between the Black and Latinx groups in this study, there was one notable difference in where participants noted obtaining health information. Black participants were more likely to emphasise obtaining

**Table 5** Example quotations of hesitancy related to distrust of institutions or concerns about structural/individual racism

| Subtheme | Quotations |
|---|---|
| Limited trust in physicians | *"I never really trust one opinion regarding health issues. I listen to what the doctors say and suggest for any illness. Next I read all information given and search the internet for reliable sources and try to gain an understanding of the situation. At that point my decision is made."*<br>*"I don't view my doctor as an expert in vaccine… Kinda like having a degree in general studies vs a specialist… I think he's knowledgeable… but don't think the level of focus and concentration points to expert."*<br>*"I don't think anyone is an expert. You can't know everything about such a new vaccine!"*<br>*"In addition to the advice of medical professionals, I also believe firmly… in the power of being natural and how people used to cure themselves in the past… The traditional remedies work."* |
| Lack of trust in government | *"I have a hard time trusting anything government affiliated—because they follow government directives rather than their own expertise."*<br>*"I don't have confidence in what the government says in general. At the end of the day they are protecting themselves and I don't believe that they are concerned about those in the lowest classes. I feel like the government if [sic] capable of lying for its own benefit."* |
| Limited trust in public health authorities | *"I trust most of their opinions. Not all."*<br>*"I trust but may not do 100% of what they say."* |
| Concerns about health outcomes differing by race or ethnicity | *"I'm not convinced that being Black does affect the risks of getting COVID. I know that's what reported but I'm just not convinced that it's true…. It's not the news itself that's unbelievable, it's the source. Medical institutions have subjected Black people to abuse, exploitation and experimentation since this country's foundation. It wouldn't be the first time that Black people were misled into getting vaccines with the false hope of immunity from a deadly disease."*<br>*"I feel that my community is more at risk of catching COVID due to the history of us being ignored by health professionals and the government. Additionally, we are most likely in employment opportunities that expose us to conditions that are not ideal. I don't agree that we are experiencing more serious reactions because that implies that we are unhealthy. Unhealthy behaviors are common in America and not assigned to simply one community. If we are having serious reactions, it is most likely due to our concerns being brushed aside when we seek assistance from health care workers."*<br>*"I honestly believe that the social structure of how Black people are treated in America is more so to do with the severity of the virus to this group. Less readily available access to health care, poor living situation, less money funneled into Black community…."*<br>*"I don't feel like my race affects my risk of getting [COVID] but I feel like it would affect the medical care that I received if I needed medical care while I was positive."*<br>*"I don't think it affects people differently due to ethnicity."*<br>*"I don't think that race is a factor here. Anyone can get the virus."* |

information about COVID-19 vaccines from the internet, despite having what appears to be strong relationships with medical providers. Latinx participants also had strong relationships with and trust in medical providers and seemed to make less use of the internet for health information. Both groups rely heavily on trusted friends and relatives for health information. Even with that support, however, moving to vaccine acceptance for some people can be very difficult and take more time than for others. As one Black participant noted: "*I love and trust my family; I love and trust my pastor. And they all made their position known. And I know none of them do things haphazardly… [but] the jury is still out for me… I'm just straddling the fence, and it's just a personal thing with me… They've all endorsed it, my pastor endorsed it…but I'm just not there, I'm not.*"

Aside from the difference noted above, we did not observe distinct themes specific to any particular subgroup (eg, by country of origin). Additionally, it should be noted that participants did not endorse conspiracy theories or unsubstantiated notions about vaccines (eg, that they contain microchips) that have been voiced in antivaccine channels.

## DISCUSSION

The results of this research suggest that interrelated barriers to vaccination are at work in communities of colour and strongly influence COVID-19 vaccine behaviours in these communities. Two main sets of concerns emerged from in these bulletin boards: that the vaccines are unsafe and that they are insufficiently effective. These concerns are remarkably similar to those observed in an earlier bulletin board study that involved a group of participants that had a majority of white people.[53] Indeed, these may be ubiquitous influences on vaccine hesitancy across racial, ethnic and national groups.[54–56]

Several participants seemed eager to make clear they felt race and ethnicity were factors in community viral infection susceptibility because of the history of structural racism in healthcare and medicine. Lack of access

to healthcare and to vaccination sites has been found to be a factor in limiting vaccination among Black and Latinx people.[37] The history of racism and medical experimentation on people of colour in the USA was cited as among the reasons for vaccine scepticism among Black participants in one recent study.[57] However, as in the survey study of Martin *et al*,[23] participants in our study did not frequently express a conviction that historical racism was a factor in their personal decisions about vaccination. Two of the 30 participants in our Black participant group directly named the Tuskegee experiments, a hallmark of unethical, racist scientific and healthcare practices in the USA. Thus, although participants in both groups often cited examples of structural racism in general, they were more likely to express individual feelings of fear and scepticism about the vaccines as the main factors in making them hesitant to be vaccinated.

Historical traumas such as the experiments that took place in Tuskegee may still have an effect on people's attitudes and decision-making even if not explicit.[58 59] Current experiences with racism such as health outcome disparities may be as or even more important in shaping ways that people of colour make decisions about healthcare issues like vaccination.[60] It is possible that the way we framed questions in these bulletin boards influenced participants towards speaking more about their individual concerns as the main factors in COVID-19 vaccine decision-making and away from broader discussions about the impact of historical and present racism on those decisions. It is also possible that for these participants at least, while recognising that structural factors like crowded work conditions and lack of healthcare access make communities of colour more likely to acquire COVID-19 and to have more negative outcomes, individual fears and scepticism about the COVID-19 vaccines were indeed the most pressing concerns that influenced vaccine hesitancy.

Although participants expressed trust in their own personal healthcare providers, they exhibited a general lack of trust in agencies and institutions that are charged with the responsibility of informing and reassuring the public about vaccine safety and efficacy. This is not a unique finding. Previous studies have found people in Black and Latinx communities have low levels of trust in the healthcare system,[27] and racial differences in healthcare access has been noted as a contributing factor.[28]

This study also raises key questions about the information environments participants were immersed in. In his Special Advisory on misinformation in 2021, the US Surgeon General discussed the need to build a healthy 'information environment'.[61] While a worthy endeavour, there are currently no standard ways to measure whether individual or community is immersed in a healthy information environment, defined as where people and communities are immersed in high-quality information of public health importance and enveloped by a communication context that underscores the trustworthiness and importance of that quality.[62] It is important to note that, at the time the bulletin boards were being done, media coverage

of about the vaccines contained a mixture of concerning and reassuring information, first about breakthrough infections, viral variants and prominent reporting on rare vaccine adverse side effects, and second, describing a public health and scientific consensus that the vaccines were safe and they were highly effective at preventing morbidity and mortality due to COVID-19.

Yet quotations from participants generally reflected more of the concerns than reassuring information, providing some insight into what their information environment may consist of. Specifically, while it was clear that participants saw and heard abundant information about vaccines, much of that information did not appear to accurately present information about both individual risks and collective benefits. Concerningly, suggests that the concerns that were circulating in these communities then met most definitions of scientific misinformation.[63 64] An alternative explanation is that despite the presentation of high-quality information, misinformation or worrisome information about the vaccines could have been more effective at shaping vaccine-related feelings and decisions. In either case, the implications are concerning. Therefore, it seems unlikely that merely supplying more facts about vaccine safety and efficacy will be sufficient on its own to sufficiently modify that information environment and change participants' views on obtaining vaccines.

Some key points that public health professionals may consider when contemplating how to encourage vaccine uptake in their work with marginalised communities include:

► Leveraging trusted sources to challenge narratives of safety and inefficacy by emphasising personal and communal benefits over risk (eg, to protect one from new strains or to protect loved ones, eg,). While previous studies of HPV vaccines observed Black people did not leverage family and friends for information,[65 66] our data suggest this may differ by vaccine. Given how both Black and Latinx participants noted obtaining much of their health information from family and friends, encouraging those who have been vaccinated to reassure unvaccinated family members and friends about safety may be an efficient way to disrupt these narratives in these priority communities. This approach would be consistent with others' recommendations to strengthen pro-vaccine messages to leverage non-medical, in-group spokespeople to share community benefits of vaccines with Black communities.[67]

► Explicitly and more emphatically framing both risks of COVID-19 and benefits of vaccination in community-level terms. It is unclear why participants often offered narratives of racial discrimination at structural and community levels yet narratives about information gathering, the negative effects of COVID-19 and benefits from vaccination were not conceptualised in a similar light. This suggests both the success of current messaging on issues of structural racism, and

the insufficiency of public health messaging to penetrate dominant media narratives framing aspects of COVID-19 in primarily individualistic terms.[68] In this way, the data presented here reflect concerns stated elsewhere about the 'individualisation of pandemic control'.[69 70]

► As many participants noted institutional mistrust in government and the health system, it will be essential for public health professionals to leverage partnerships to effectively reach marginalised community members with trusted messengers. This includes improving access to vaccines for people in traditionally underserved communities as access issues can foment mistrust and suspicion of the healthcare system and efforts to ameliorate them may make some people more likely to accept vaccines.[71] Access must also be coupled with sufficient training to primary care clinicians to build trust with Black and Latinx communities. As many participants noted that they did not consider their physician sufficiently expert to trust their opinion on vaccines, counteracting misinformed ideas held about vaccines by patients will only be successful if there is sufficient trust in the relative expertise of those care providers.

## Limitations

This study has several limitations, including inherent self-selection bias in the sample of participants. There is also inevitable bias towards the views of those comfortable sharing their opinions in a group discussion with others in a digital setting where social desirability bias may make some participants reticent to share what may be perceived as outlandish opinions. This may have been a factor in the fact that subjects did not, for the most part, mention historically racist events and the US racialised history does not rule out that these are important factors for vaccine hesitancy. We did not ask specific questions about these issues. Participants' reports of mistrust of public health authorities and the government represent the result of both historical racism and personal experiences of racism. Thus, while we can report our observation that for the most part neither people in our Black nor Latinx groups volunteered racism as affecting their own vaccine decisions, deeper probing might have elicited that as an important factor. Indeed, Dong *et al* conducted semistructured interviews with 24 Black Americans and reported that, 'systemic racism was discussed as the root cause of the different types of mistrust'.[72] Finally, while we initially asked questions about influenza, these discussions yielded surprisingly narrow themes. We suspect but cannot confirm that this observation was connected to of the context and timing of our study as data was collected in the first months after the release of COVID-19 vaccines.

In summary, bulletin boards with COVID-19 vaccine-hesitant people from the Black and Latinx communities revealed that the major factors influencing vaccine hesitancy involve fears of lack of safety and efficacy of the vaccines. There is a misperception that not being vaccinated is a social norm because of media emphasis on unvaccinated people. These attitudes are reinforced by a perception of lack of consensus about the vaccines among experts, mistrust of government officials and institutions, and belief that other measures are sufficient to prevent acquisition and spread of COVID-19. Future research will focus on strategies to improve vaccine acceptance that do not rely only on providing facts but account also for the anxieties and fears that motivate vaccine hesitancy.

**Acknowledgements** This work was supported by the Robert Wood Johnson Foundation (grant numbers 76935 and 78084) and Weill Cornell Medicine's JumpStart program (grant number N/A). We also thank the peer reviewers for their thoughtful and insightful comments.

**Contributors** DS and JMG conceived of the study, wrote the protocol, obtained funding and engaged WS and NG to carry out the research and conduct preliminary data analysis. DS, SG, JMG, WS and NG were responsible for inclusion/exclusion criteria of both bulletin board groups. WS and NG were responsible for bulletin board design, and recruitment of participants. WG undertook initial interpretive phenomenological and narrative analysis. SG, SW, LH, MR and AA contributed to further data review and interpretation with DS and JMG. DS is responsible for the overall content of the work, had access to the data and controlled the deicision to publish. All authors read and critically evaluated multiple drafts of the manuscript before providing final approval of the version to be published.

**Funding** This work was supported by the Robert Wood Johnson Foundation (grant numbers 76935 and 78084) and Weill Cornell Medicine's JumpStart program (Grant number N/A).

**Disclaimer** The opinions expressed in this piece do not necessarily reflect those of either the Robert Wood Johnson Foundation or Weill Cornell Medicine.

**Competing interests** None declared.

**Patient and public involvement** Patients and/or the public were not involved in the design, or conduct, or reporting, or dissemination plans of this research.

**Patient consent for publication** Not applicable.

**Ethics approval** This research was deemed exempt from IRB review by Ethical and Independent Review Services and approved by the Weill Cornell IRB (19-10020908).

**Provenance and peer review** Not commissioned; externally peer reviewed.

**Data availability statement** Data are available on reasonable request. Due to privacy concerns, data are not currently publicly available, but deidentitified data can be obtained by researchers on a case-by-case basis by contacting the authors. Patients were not involved in the design of the study.

**ORCID iD**
David Scales http://orcid.org/0000-0001-5727-7148

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
