## [Reviewer comments · BMJ Open]

ARTICLE DETAILS

TITLE (PROVISIONAL)	"They've all endorsed it...but I'm just not there:" A qualitative exploration of Covid-19 vaccine hesitancy reported by Black and Latinx individuals
AUTHORS	Scales, David; Gorman, Sara; Windham, Savannah; Sandy, William; Gregorian, Nellie; Hurth, Lindsay; Radhakrishnan, Malavika; Akunne, Azubuike; Gorman, Jack M.

VERSION 1 – REVIEW

REVIEWER	Meyer, Samantha University of Waterloo
REVIEW RETURNED	09-Mar-2023

GENERAL COMMENTS	This is an important study that has potential to make novel contributions to our understanding of hesitancy among Black and Latinx communities in the United States. However, I have many questions/concerns that could be addressed before it might be suitable for publication. I hope that in reviewing and perhaps responding to these items, the manuscript makes a more evident and sound contribution to the literature. Abstract: I suggest the objective be reworded as I don't agree that you can "examine an in-depth analysis" unless you are looking at the analysis of data collected by others. Page 4, line 6 - Typo – reasons (not resons) With regards to the objective, was the data viewed through a lens of "historical racism, present-day structural racism, medical distrust, and individual concerns about vaccine safety and efficacy", was this a prediction (akin to a hypothesis) or was this a line of inquiry that drove the development of data collection tools? Introduction: -It would be helpful to readers if the team might specify the social determinants, drawing on literature and references to support their statement. It's not that I challenge the statement but it would be helpful to root such statements in evidence. Additionally, what is meant by 'substantial' in the following sentence "a substantial number of people continue to choose not to be vaccinated"? -The background content in terms of topics included is important but is also too vague at times. I suggest the authors provide grater engagement with the specifics of the literature that led them to generate their objective. -How do the authors define/conceptualize vaccine hesitancy? -Does mistrust differ from distrust?
---

	-When referring to the sample, is this inclusive of immigrants and refugees as well? It will be important to consider the heterogeneous nature of Black and Latinx communities. Were the Latinx participants recruited living in the US, or the countries named in the methods? -While I agree with this statement “Additionally, healthcare access disparities remain an important issue that intersects with manifestations of medical racism within the healthcare system”, the literature that follows does not directly support it, and there is no reference cited. -Page 7, line 14 – while I agree with the three broad factors that influence vaccine decisions that form the objective of the paper, I think the background literature could be organized to highlight these better, and a stronger argument could be made (drawing on literature or by editing) to justify the three areas named. Page 7 line 24 – uptake, not update -The paper could use another read to catch typos and some areas where wording distracts from clarity of arguments. -The authors note that participants were sampled to match the intervention group – how did they manage this and what did the intervention group look like? Is there any reason the same groups weren’t used for both studies? -Did you define VH for participants? I would argue that people who had already been vaccinated or intended to be vaccinated soon could also be vaccine hesitant. You might consider referring to your panel as individuals who did not plan to get vaccinated but did not identify as rejecting of vaccines – something to that effect. Providing your definition of vaccine hesitancy might also help. -Can the authors provide a date (rather than ‘soon after vaccine availability’) as different vaccines were rolled out at different times, eliciting different responses. -Please describe in a bit more detail the use of ‘bulletin boards’. Are these run by private companies or on social media? Who are the participants (if there are data on this) and how did you recruit for this specific study? Did you enter an existing bulletin board or are these set up to answer research questions? Why was ethics considered unnecessary? I have heard of cases where ethics are not considered necessary in the use of social media data that already exist and researchers are instead asking questions of available data. However, from my read it was primary data collected on these board. The posts were used to elicit responses, correct? I’d assume that any line of questioning would need to be approved. Is the moderator always a researcher? -How did you identify the panel of people who agreed to be surveyed? -Why were questions about influenza asked? -It is suggested that tables be listed consecutively – with 1 and then 2. -Can the actual questions posted on the boards be provided? There is a long list of topics provided. Were each of these one question and were responses given by all participants to each topic? Did the answers on day 1 inform questions on subsequent days? Were the same topics used in both population group boards? -It is stated that sampling was purposive for diversity – were any themes distinct for specific groups within the two populations? -Was content analysis conducted, as stated? Can the authors please provide details on why this was done and what it offered as distinct from the other approaches to analysis in terms of meeting the objective?
--	--

	-It would be helpful as a reader to see the quotes within the result section rather than looking at the themes and confirming with the data. I know this might be a function of formatting for submission but it was cumbersome to go between the two. -It wasn't clear to me in the results how IPA and narrative analysis were used. It reads to me like a thematic analysis. Can the authors please comment on the presentation of data and how this speaks to narratives and/or experiences? -I question the use of the word 'convince' in reference to vaccine acceptance, especially among populations with low/no trust in government institutions. I would suggest reconsidering our role as promoting vaccination, and how we might do this working with communities. -Please ensure that the discussion points/arguments made are data driven. The literature should be used to discuss the results, but conclusions drawn should be rooted in data. The list of bullet points in this section, for example, could be revised to better present the take home messages from the data and engage with more meaningful literature and discussion.
--	---

REVIEWER	Cáceres, Nenette A. Cedars-Sinai Medical Center
REVIEW RETURNED	19-Mar-2023

GENERAL COMMENTS	This qualitative study examined reasons for vaccine hesitancy among online communities of US-based Black and Latinx communities to understand the role of historical racism, present-day structural racism, medical distrust, and individual concerns about vaccine safety and efficacy. This study was very interesting and informative. The combination of qualitative methodologies was strong but can be improved. I suggest adding information on the research paradigm utilized and technique used to assess credibility of data analysis. Lastly, include details on what software was used for data analysis (if any) and/or how coding was done.
--

VERSION 1 – AUTHOR RESPONSE

Reviewer: 1

Dr. Samantha Meyer, University of Waterloo

Comments to the Author:

Comment:

This is an important study that has potential to make novel contributions to our understanding of hesitancy among Black and Latinx communities in the United States. However, I have many questions/concerns that could be addressed before it might be suitable for publication. I hope that in reviewing and perhaps responding to these items, the manuscript makes a more evident and sound contribution to the literature.

Response: We are grateful for Dr. Meyer's thoughtful review of our manuscript. We appreciate the suggestions and believe it has made the manuscript clearer and more useful for potential readers.

Comment:

Abstract:

I suggest the objective be reworded as I don't agree that you can "examine an in-depth analysis" unless you are looking at the analysis of data collected by others.

Response: Thank you for pointing this out. We have removed the reference to an "in-depth analysis." The term also appeared in the penultimate paragraph of the introduction so we have removed that reference as well.

Comment:

Page 4, line 6 - Typo – reasons (not resons)

Response: We apologize for this error and have corrected it.

Comment:

With regards to the objective, was the data viewed through a lens of "historical racism, present-day structural racism, medical distrust, and individual concerns about vaccine safety and efficacy", was this a prediction (akin to a hypothesis) or was this a line of inquiry that drove the development of data collection tools?

Response: Thank you for this opportunity to clarify. This was not a hypothesis but part of our reflective process, noting various factors talked about in the literature and popular media on vaccine hesitancy that therefore formed the basis for the bulletin board facilitator guide and an acknowledgement that we anticipated participant perspectives to be framed along these various lines. However, while we wanted to understand the role of these factors, the research design and methodology was chosen to help us gather information that could allow us to see past these concepts, or, at the very least, provide insight on how they manifest from participants' perspectives.

Comment:

Introduction:

-It would be helpful to readers if the team might specify the social determinants, drawing on literature and references to support their statement. It's not that I challenge the statement but it would be helpful to root such statements in evidence.

Response: Thank you for this helpful comment, and the related comments below suggesting a way to reorganize the introduction to provide clarity for the reader. In this revision we have done the following:

- Reorganized the introduction as per your thoughtful suggestion along the lines of the topics noted in the objectives of the paper

- Included specific mention of particular structural and social determinants of health that impact racial disparities in vaccination. Though here we note that the paper by Njoku et al. 2021 is quite comprehensive, so we did not list all of the SDoH detailed in that excellent paper.
- Included more literature specific to the objectives of the paper, though we recognize that this literature is vast and our overview not comprehensive. We therefore welcome any specific citations the reviewer perceives to be of particularly high value.

Comment:

Additionally, what is meant by 'substantial' in the following sentence "a substantial number of people continue to choose not to be vaccinated"?

Response: The word "substantial" was intentionally chosen for its vagueness. As this proportion changes over time, we sought to avoid a specific number that would be out of date by the time of publication, but to put emphasis on the fact that a portion of the US population, especially among marginalized communities noted in the citations, remains unvaccinated.

While we have purposely chosen vagueness to help emphasize the overall persistence of a group that has not been vaccinated, we recognize the subjective nature of the word "substantial" and have removed it.

Comment:

-The background content in terms of topics included is important but is also too vague at times. I suggest the authors provide greater [sic] engagement with the specifics of the literature that led them to generate their objective.

Response: Please see the response above regarding the reorganization of the introduction.

Comment:

-How do the authors define/conceptualize vaccine hesitancy?

Response: We have added the following definition to the text: "indecision around accepting a vaccination," but note in the methods that for practical purposes we used a more specific definition for patient recruitment.

Comment:

-Does mistrust differ from distrust?

Response: Thank you for raising this point. While the terms were used interchangeably by participants, we recognize the difference (drawing from Barbalet 2009) and have 1. Adjusted our language in the paper to ensure clarity, and 2. Double checked to ensure that any participant quotations did not contain language that may be similarly confusing.

Comment:

-When referring to the sample, is this inclusive of immigrants and refugees as well? It will be important to consider the heterogeneous nature of Black and Latinx communities. Were the Latinx participants recruited living in the US, or the countries named in the methods?

Response: As noted in Table 1, the sample is inclusive of immigrants and refugees as the main criteria was residency in a US state or territory given the funder's priority on US-based populations. We did this purposely to ensure including various sub-communities within Black and Latinx groups. Our reason for asking about country of origin was to understand if we had a diverse sample.

Comment:

-While I agree with this statement "Additionally, healthcare access disparities remain an important issue that intersects with manifestations of medical racism within the healthcare system", the literature that follows does not directly support it, and there is no reference cited.

Response: This statement has now been removed in the reorganization of the introduction noted above.

Comment:

-Page 7, line 14 – while I agree with the three broad factors that influence vaccine decisions that form the objective of the paper, I think the background literature could be organized to highlight these better, and a stronger argument could be made (drawing on literature or by editing) to justify the three areas named.

Response: Please see the response above regarding the reorganization of the introduction.

Comment:

Page 7 line 24 – uptake, not update

Response: Thank you for pointing out this misstatement. It has been corrected.

Comment:

-The paper could use another read to catch typos and some areas where wording distracts from clarity of arguments.

Response: Thank you for this suggestion and we apologize for these oversights. After revisions were done, the manuscript was reviewed by co-authors, including a dedicated “copy-edit” readthrough prior to resubmission.

Comment:

-The authors note that participants were sampled to match the intervention group – how did they manage this and what did the intervention group look like? Is there any reason the same groups weren't used for both studies?

Response: Thank you for this helpful observation that may be confusing to readers. Our interventions are in organic online spaces, meaning these are public environments where real people are interacting in real time. Demographic data on these public groups does not exist, nor is it feasible to undertake it. Instead, our interventionists observed these environments and examined public profiles of those participating to get an approximate sense of the demographic characteristics of the population interacting in these digital spaces, trending slightly toward younger, female audiences in the Latinx group and younger but approximately gender equal in the Black group. Other demographic characteristics that cannot be gleaned from observation of public social media profiles (e.g. marital status, education level or household income) could not be assessed so were not accounted for in recruitment but were included in Table 1 for completeness. We have clarified this in the manuscript by adding the phrase “based on observational assessment of such spaces by our interventionists” and adding citations to our work on these interventions, though it should be noted that while the works cited outline the intervention process and context, the manuscript that deals specifically with interventions in Black and Latinx online spaces that drew from these bulletin boards is currently under development.

Comment:

-Did you define VH for participants? I would argue that people who had already been vaccinated or intended to be vaccinated soon could also be vaccine hesitant. You might consider referring to your panel as individuals who did not plan to get vaccinated but did not identify as rejecting of vaccines – something to that effect. Providing your definition of vaccine hesitancy might also help.

Response: We did not provide potential participants with a definition of vaccine hesitancy. For recruiting purposes, we created our own definition of vaccine hesitancy -- as adults who were not categorically opposed to vaccines but were undecided as to whether it was safe to receive the new COVID-19 vaccines. We then screened for individuals that met these criteria. For the reviewers' convenience we have included the specific questions used to screen potential participants below. We have also noted our definition in the methods section of the paper. While the reviewer makes a good point that those who had been vaccinated or intended to be vaccinated soon could also have been vaccine hesitant, our focus was on understanding the rationale for why people were not getting vaccinated. We therefore considered those that were hesitant but still chose to receive the vaccine in a different category.

- 10) Which statement best describes your attitude towards vaccines in general – not just flu vaccines?
 Vaccines are generally important and safe if they are approved by public health experts.
 Most vaccines are safe and important to have but some are not safe or not important to have.
 Most vaccines are unsafe or unnecessary. **TERMINATE**
 All vaccines are unsafe or unnecessary. **TERMINATE**
 I don't know or don't have an opinion on whether vaccines are safe or necessary.
- 11) If you have children under the age of 13, have any of them received any vaccines?
 Yes, at least one of my children under the age of 13 has received a vaccine
 No, none of my children under the age of 13 have ever received a vaccine
 I do not have any children under the age of 13
- 12) **[IF YES]:** Have you ever declined a vaccination for your child because you believed that it was unsafe or unnecessary?
 Yes **TERMINATE**
 No
 No, but I have requested to have my child's vaccinations delayed or spaced out
- 13) **[IF NO CHILDREN UNDER 13]:** If you had an infant or toddler, would you allow them to receive vaccines recommended by their pediatrician?
 Yes, I would allow my child to receive all vaccines recommended by their pediatrician.
 Yes, I would allow my child to receive some vaccines recommended by their pediatrician, but not all.
 No, I would not allow my child to receive any of the vaccines recommended by their pediatrician. **TERMINATE**
 I don't know
- 15) When the influenza vaccine – often called the “flu” vaccine -- is made available each year, do you...
 Usually get it
 Sometimes get it
 Never get it
RECRUIT A MIX
- 16) What is your intention regarding a COVID vaccine?
 I have already received a COVID vaccine. **TERMINATE**
 I intend to get a COVID vaccine in the near future. **TERMINATE**
 I am undecided or hesitant about whether to get a COVID vaccine.
 I will probably get it eventually but want to wait
 I will definitely not get a COVID vaccine. **TERMINATE**
- 17) Which of the following statements best describes your attitude towards a COVID vaccine?
ACCEPT MULTIPLE
 It is important that as many people as possible get the COVID vaccine in order to minimize its spread.
 It is important that as many people as possible get COVID to develop immunity and help us reach herd immunity.
 The COVID vaccine is important to prevent me from getting sick with COVID
 The COVID vaccine is important only for people who have high risk of getting seriously ill (those who have a pre-existing condition or are very prone to getting sick).
 I don't think that catching COVID is all that bad.
 I'm not convinced that COVID vaccines work.
 The COVID vaccine is unnecessary because the virus is not real (is a hoax).
 I have serious concerns regarding the safety of the flu vaccine.

- 18) In the past year, have you heard any news, information or discussions in your community that have made you question the safety of the covid vaccine?
- () Yes, a lot
 - () Yes, some
 - () No **TERMINATE**
 - () Don't remember **TERMINATE**

Comment:

-Can the authors provide a date (rather than 'soon after vaccine availability') as different vaccines were rolled out at different times, eliciting different responses.

Response: We appreciate the reviewer's preference to have specific dates and refer the reviewer to the following paragraph (the first paragraph of the methods) in which exact dates are given. In the introduction, however, we chose to emphasize the context in which the bulletin boards were done, especially relative to vaccine roll-out, rather than highlight details that may detract from understanding that larger context. In case the reviewer was seeking more temporal specificity, we adjusted the phrasing (while trying to avoid redundancy) to say that the bulletin boards "were conducted within the first few months after Covid-19 vaccines were made available..."

Comment:

-Please describe in a bit more detail the use of 'bulletin boards'. Are these run by private companies or on social media? Who are the participants (if there are data on this) and how did you recruit for this specific study? Did you enter an existing bulletin board or are these set up to answer research questions?

Response: To answer the reviewer's questions, we have added information on the private company through which the bulletin boards are run. QualBoard (since acquired by Sago) has proprietary software for bulletin boards for research purposes and a database of people who have agreed to participate in prior studies. bulletin board was set up exclusively for this purpose – it did not exist prior to the study, and it was shut down as the study ended. As it is a private company, we do not have data on other participants that use their platform.

As we note in the Data Collection section, the participants in the Black bulletin boards were recruited from a panel of people who had previously agreed to participate in online services. They were contacted by email with an invitation to participate, and if they expressed interest, they were screened for eligibility. The participants in the Latinx communities were recruited through invitations in Spanish on various Facebook pages built for Latinx sub-populations by country of origin. Those that expressed interest were then screened for eligibility.

Comment:

Why was ethics considered unnecessary? I have heard of cases where ethics are not considered necessary in the use of social media data that already exist and researchers are instead asking questions of available data. However, from my read it was primary data collected on these boards. The posts were used to elicit responses, correct? I'd assume that any line of questioning would need to be approved. Is the moderator always a researcher?

Response: We apologize for any misunderstanding here. We do not state in the paper that ethics was considered unnecessary, though if the reviewer could point us to the section of the paper where this impression was given we would like to change it. Because we agree with the reviewer that primary data was collected and ethical oversight required and obtained. As we state elsewhere in the paper, the research was deemed exempt from IRB review by Ethical and Independent Review Services and approved by the Weill Cornell IRB (19-10020908).

Comment:

-How did you identify the panel of people who agreed to be surveyed?

Response: As noted above and in the Data Collection section, the participants in the Black bulletin boards were recruited from a panel of people who had previously agreed to participate in online services. They were contacted by email with an invitation to participate, and if they expressed interest, they were screened for eligibility. The participants in the Latinx communities were recruited through invitations in Spanish on various Facebook pages built for Latinx sub-populations by country of origin. Those that expressed interest were then screened for eligibility.

Comment:

-Why were questions about influenza asked?

Response: The funder was initially interested in this topic so focus groups were initially designed with this in mind.

Comment:

-It is suggested that tables be listed consecutively – with 1 and then 2.

Response: Thank you for this comment. We agree that consecutive numbering of the tables would be most helpful for clarity and to assist the reader. We believe we achieved that in the manuscript with the first reference to Table 1 being in the first paragraph under the section Data collection, followed by a mention of Table 2 in the last paragraph of that section. We mention Table 1 again in the first paragraph of the following section, so this may seem confusing to the reader. As demographic characteristics are typically Table 1 in many papers, we have kept this ordering but called more attention to the first time we mention Table 1 to ensure the reader does not perceive that the numbering is not consecutive.

Comment:

-Can the actual questions posted on the boards be provided? There is a long list of topics provided. Were each of these one question and were responses given by all participants to each topic? Did the answers on day 1 inform questions on subsequent days? Were the same topics used in both population group boards?

Response: We have now included the facilitator guide as part of the supplementary material. The reviewer will note that there was a mixed structure to the questions, sometimes one question, sometimes a question with a follow up and request for explanation (e.g. “How would you describe your level of trust with your personal physician? Why is that? Please explain your answer.”). While day 1 answers did not inform questions on subsequent days because the facilitator guide was drawn up in advance, the facilitator was the same throughout all days of the bulletin boards and leveraged what people had said earlier in the discussion to ask further follow-up questions for clarification, noting contradictions, etc, similar to how a focus-group facilitator may function. The facilitator guide was translated into Spanish and used for the Latinx bulletin boards as well.

Comment:

-It is stated that sampling was purposive for diversity – were any themes distinct for specific groups within the two populations?

Response: Aside from the differences already noted between the two bulletin boards, we did not observe distinct themes specific to any particular sub-group (e.g. country of origin). We note, however, that our sample size was small, so the purpose of the diverse sampling was not to find or characterize distinct themes specific to a particular sub-population, but to ensure a range of viewpoints both to trigger discussion within the bulletin board as well as to broaden our analysis. We have noted this in the “other observations” section of the Results.

Comment:

-Was content analysis conducted, as stated? Can the authors please provide details on why this was done and what it offered as distinct from the other approaches to analysis in terms of meeting the objective?

Response: We appreciate the reviewer raising this point. Content analysis and thematic analysis are often seen as being at two ends of a spectrum of qualitative analysis, so we can see how it may be confusing that we brought both types of methods to bear on our data. We have clarified that we employed directed content analysis as a secondary form of analysis for an additional check to ensure we met the objectives of the study. We have also added a citation to Hsieh et al. 2005, which provides more in-depth reasoning on this type of secondary analysis approach using directed content analysis.

Comment:

-It would be helpful as a reader to see the quotes within the result section rather than looking at the themes and confirming with the data. I know this might be a function of formatting for submission but it

was cumbersome to go between the two.

Response: We understand that it can be cumbersome to move between a table and the text and we agree that this is likely a function of formatting for submission. The benefit of a table (in the final published format) is having all the data in one place, which some readers prefer (as do we). We apologize that this current formatting is not conducive to convenient reading and appreciate the reviewer's thoughtful comments despite this challenge.

Comment:

-It wasn't clear to me in the results how IPA and narrative analysis were used. It reads to me like a thematic analysis. Can the authors please comment on the presentation of data and how this speaks to narratives and/or experiences?

Response: Thank you for raising this point. We agree that the structure and presentation of the data is consistent with a thematic analysis and we hope to provide a better explanation here. Our reasoning for the multiple analysis methods was to increase credibility and dependability; however, we quickly recognized that this presents a challenge in presenting the results. To avoid redundancy, we chose to present results in a way that would be both parsimonious and intuitive for most reviewers to follow. We found that IPA most useful in framing the overall experiences participants had as they considered their decisions about vaccines, manifesting in the broad themes around which the tables were constructed. The narrative analysis was most useful in understanding participants' understandings and sensemaking of how systemic or individual racism factored into the COVID-19 situation, either around higher mortality rates, care disparities, or disparities in vaccine demand.

Perhaps we have been too parsimonious in not explicitly noting the points where IPA and narrative analysis brought out interesting insights from the data, but we were unsuccessful in finding ways to do this that did not recapitulate quotations presented elsewhere. Our approach was therefore to present the data in an organized format with abundant examples, then draw on those examples in our discussion to highlight points raised through narrative and phenomenological aspects.

The tensions created by this approach were perhaps picked up by the reviewer in this comment and in comment below about the discussion points being data driven. As with most qualitative research, what we have presented here does not reflect our entire analysis, which contained much more detail about sensemaking around vaccines, positionality and identity based on participants' race or ethnicity, and also their feelings about influenza vaccines. Here we sought to present a selected portion of the results focused on the objectives of the study. To address the reviewer's concerns, we have sought to change the language in the discussion section to highlight this sensemaking and predominant narratives.

Comment:

-I question the use of the word 'convince' in reference to vaccine acceptance, especially among populations with low/no trust in government institutions. I would suggest reconsidering our role as promoting vaccination, and how we might do this working with communities.

Response: Thank you for raising this point. We appreciate the implications of "convincing" vaccine hesitant people and have rephrased the statement as follows: "may be more useful as public health

professionals reconsider how to encourage vaccine uptake in their work with marginalized communities”

Comment:

-Please ensure that the discussion points/arguments made are data driven. The literature should be used to discuss the results, but conclusions drawn should be rooted in data. The list of bullet points in this section, for example, could be revised to better present the take home messages from the data and engage with more meaningful literature and discussion.

Response: We appreciate the reviewer raising this point and have made the following adjustments to this section of the manuscript:

- We have revised these bullets to more explicitly draw from the data presented in the paper
- We have added more linkages to existing literature to buttress the arguments made
- We have removed bullets that seemed on weak foundation relative to the data presented in the paper

Reviewer: 2

Dr. Nnette A. Cáceres, Cedars-Sinai Medical Center

Comments to the Author:

Comment:

This qualitative study examined reasons for vaccine hesitancy among online communities of US-based Black and Latinx communities to understand the role of historical racism, present-day structural racism, medical distrust, and individual concerns about vaccine safety and efficacy. This study was very interesting and informative.

Response: We are grateful for Dr. Cáceres’ thoughtful review of our work.

Comment:

The combination of qualitative methodologies was strong but can be improved. I suggest adding information on the research paradigm utilized and technique used to assess credibility of data analysis.

Response: We are happy to provide more detail on this point. We have added to the methods section that we took a reflexive stance and a phenomenological constructivist approach that we believed

melded well with our primary analysis method. We also added a citation in interpretive phenomenological analysis that we found helpful (Frechette et al. 2020).

Regarding the credibility of the data, we have added the following paragraph to the methods section:

“Data credibility was assessed through discursive triangulation. Initial coding was followed by a process of layered discussions. Specifically, that meant multiple meetings between the primary data analyst and a supervisor to process codes and themes, followed by further coding and thematic discussions and revisions with the first and senior author. After consensus on themes had been reached by those four authors, subsequent review and discussion then took place with the remaining co-authors.”

Comment:

Lastly, include details on what software was used for data analysis (if any) and/or how coding was done.

Response: As the coding was done manually, there was no software used for data analysis. The following details of the manual coding process have been added to the methods section:

“Initial coding was done manually by WS. Transcripts were read through once in their entirety by prior to coding. On a second readthrough, relevant text was highlighted, codes were inductively drawn out, and labeled in text margins. Codes were then aggregated and organized into themes which were discussed in meetings with co-authors discussed in more detail below. Themes were then arranged with key quotations pulled out as illustrative examples.”

VERSION 2 – REVIEW

REVIEWER	Cáceres, Nnette A. Cedars-Sinai Medical Center
REVIEW RETURNED	21-Jun-2023
GENERAL COMMENTS	Thank you for clarifying the concerns noted during my initial review.

VERSION 2 – AUTHOR RESPONSE